# Short-term effects of national-level natural resource rents on life expectancy: A cross-country panel data analysis

**Isaac Lyatuu**[1,2,3]*, **Georg Loss**[2,3], **Andrea Farnham**[2,3], **Mirko S. Winkler**[2,3], **Günther Fink**[2,3]

**1** Ifakara Health Institute, Dar es Salaam, United Republic of Tanzania, **2** Department of Epidemiology and Public Health, Swiss Tropical and Public Health Institute, Basel, Switzerland, **3** University of Basel, Basel, Switzerland

* ilyatuu@ihi.or.tz

**Data Availability Statement:** All data files are publicly available from the World Bank Development Indicators website (URL: https://databank.worldbank.org/source/world-

## Abstract

While a substantial amount of literature addresses the relationship between natural resources and economic growth, relatively little is known regarding the relationship between natural resource endowment and health at the population level. We construct a 5-year cross-country panel to assess the impact of natural resource rents on changes in life expectancy at birth as a proxy indicator for population health during the period 1970–2015. To estimate the causal effects of interest, we use global commodity prices as instrumental variables for natural resource rent incomes in two-stage-least squares regressions. Controlling for country and year fixed effects, we show that each standard deviation increase in resource rents results in life expectancy increase of 6.72% (CI: 2.01%, 11.44%). This corresponds to approximately one additional year of life expectancy gained over five years. We find a larger positive effect of rents on life expectancy in sub-Saharan Africa (SSA) compared to other world regions. We do not find short-term effects of rents on economic growth, but show that increases in resource rents result in sizeable increases in government revenues in the short run, which likely translate into increased spending across government sectors. This suggests that natural resources can help governments finance health and other development-oriented programs needed to improve population health.

## Introduction

The extraction of natural resources such as minerals, oil and gas has the potential to drive growth, reduce poverty and promote sustainable development [1, 2]. This is particularly relevant for low- and middle-income countries in Africa and Latin America, where the mining industry might play an even more important role in a low carbon future [3–5]. Renewable energy sources and energy storage batteries are significantly more material-intensive in their composition than traditional fossil-fuel-based energy supply systems, which might result in a rapidly increasing demand in relevant metals [5–7].

A large body of literature has highlighted the negative association between natural resource endowment and economic development, often referred to as the "*resource curse*" [8–14]. The

development-indicators). There are no special access privileges that others would need to provide.

**Funding:** The authors would like to acknowledge the Swiss Programme for Research on Global Issues for Development (r4d Programme), which is a joint funding initiative by the Swiss Agency for Development and Cooperation (SDC) and the Swiss National Science Foundation (SNSF, grant no 169461). The funders had no role in study design, data collection and analysis, decision to publish, or preparation of the manuscript.'

**Competing interests:** The authors have declared that no competing interests exist.

principle mechanisms underlying the resource curse outlined in the literature are i) declines in manufacturing and agriculture sectors due to exchange rate appreciation ("Dutch disease"), ii) weakened institutions due to increased rent-seeking and corruption, iii) debt overhang resulting from excess government borrowing against natural resources, and iv) reduced enrollment into higher education due to increasing availability of mining jobs [15–18]. Despite these plausible causal pathways, empirical evidence on the resource curse remains remarkably inconsistent. A recent meta-analysis shows that approximately 40% of empirical papers find negative associations between resource endowments and economic growth, 40% find no associations, and 20% find positive links [19]. In general, the negative relationship between natural resources and economic growth seen in cross-sectional models in Sachs and Warner [13] seem to disappear when country fixed effects are introduced in empirical models [20] and when measurement error concerns are appropriately addressed [21].

One key factor likely to critically shape the nonlinear relationship between resource endowment and economic growth is institutional quality [22–25]. Countries with weak institutions may be heavily affected by natural resource-related rent-seeking behaviour and corruption [26], and may provide owners of natural resources and other elites easy access to political power [27–29]. With strong institutions, natural resources can be converted into productive assets and human capital and contribute to economic development [15].

Most of the literature on natural resource endowments has focused on economic growth. Much less is known regarding the relationship between natural resources endowment and population health. Population health is influenced both directly and indirectly through activities associated with resource extraction [30]. Even though health effects of extractive industries on population health are rarely measured [31–33], both positive and negative effects on health outcomes such as the prevalence of malnutrition, vector-related diseases, sexual transmitted infections including HIV/AIDS and mental health seem plausible [34–37]. The potential inter-linkages between population health and natural resource extractraction are of particular concern for Africa: the continent with the highes burden of diseases [38], the lowest life expectancy at birth [39] and the highest concentration of natural resources such as oil, copper, diamonds, bauxite, lithium and gold [40]. Hence, Africa's wealth in natrual resources presents both a risk and an opportunity for public health in producer regions [41].

While extractive industries may contribute to improved health outcomes locally through improved infrastructure, employment and business opportunities, these effects are likely too small to be reflected in national estimates due to the relatively small populations directly exposed to such projects. Transitory income shocks have been used to assess the impact of GDP per capita on a range of outcomes including conflicts, democracy, population growth and civil wars [42–45]. Brückner et al. [44] show that exogenous increases in international oil prices can affect countries' population growth as well as economic development. In the present paper, we add to the existing literature on the links between a broader range of natural resource endowments and countries' development by assessing the causal relationship between natural resource endowment and life expectancy at birth as the most commonly used proxy for overall population health. Given the importanc of mining for SAA, we separately present results for this region versus the rest of the developing world, and also show population health and mining trajectories for selected countries with large natural resource endowments in the subcontinent.

## Data and methods

### Data

We downloaded and combined data from two data sources: (1) World Development Indicators (WDI)–The World Bank [46]; and (2) Pink Sheet Data—The World Bank [47]. The WDI

databank provides annual data on different series of development indicators, covering the period from 1960 to 2019. We used STATA/IC 15.0 [48] to download and analyze the data. Using the STATA plugin WBOpendata [49], we downloaded annual and country-specific indicators on Gross Domestic Product (GDP), life expectancy and natural resources rents, and converted these data into a five-year panel data set. Our panel data comprises 1990 country-year observations covering 199 territories and ten 5-year intervals between 1970 to 2015. The 199 territories included the 193 United Nations member states [50], China Macao, China Hong Kong, Greenland and Kosovo, as well as West Bank and Gaza as special administrative regions. Data availability for the period before 1970 was extremely limited in the WDI database. The same was also true for recent years and therefore limited our analysis to 1970–2015, with a total of 186 territories contributing to the core analysis.

## Additional calculated indicators

We derived the country's institutional quality (IQ) from the mean value of the three IQ indicators (rule of law, government effectiveness and control for corruption). We obtained country average IQ and country average current health expenditure (CHE) by averaging non-missing values for a given country using a full study sample. We then assigned countries to either low or high IQ and either low or high CHE based on these average values.

## Exposure variable

Our independent variable of interest is country's total natural resource rents. We follow most of the literature in using the percent share of GDP as our primary measure for total natural resources rents. The main exception from this approach are papers by Sachs & Warner [11, 51], who measured resource abundance as the share of primary-product export over GDP. The main problem with this measure is that it does not account for nonrenewable products such as gold and diamonds, which account for a significant proportion of exports in resource rich countries [22]; it is also heavily affected by the overall composition of each country's export sector [23]. Total natural resource rents are the sum of oil, natural gas, coal, mineral, and forest rents. Individual rents are calculated by taking the difference between the price and the production cost of each commodity and multiplying this margin by the total quantity of the specific commodity extracted [46]. Additional covariates are (1) HIV prevalence, (2) percentage of urban population, (3) secondary school enrollment, (4) tertiary school enrollment, (5) total government revenue, (6) rule of law, governance effectiveness and control for corruption, (7) CHE and (8) foreign direct investment. Definitions of these variables can be found in S1 Appendix.

## Global price series data

Data on global price series were extracted from World Bank Commodity Price Data ("The Pink Sheet") in nominal, as well as in real 2010 USD. The Pink Sheet contains commodity price indices on major commodities grouped as energy (crude oil, natural gas, coal), non-energy (cocoa, coffee, tea), agricultural, fertilizers, metals and minerals (aluminium, copper and iron), and precious metals (gold and silver). Further details on these indices are provided at the World Bank website [47].

## Missing data and imputation

To address missing data in the WDI database, we used Multiple Imputation Chained Equation (MICE) with Predictive Mean Matching, ten nearest neighbours and fifty iterations on the additional set of variables.

## Statistical methods

### Empirical strategy

To overcome both confounding and measurement error concerns in cross-national analysis, we use exogenous variations in global commodity prices as instrument to estimate local average treatment effects (LATE) in an instrumental variable (IV) regression model. We calculate country's predicted rents by multiplying 5-years averages of country's total rents and the five energy indices. In order to illustrate the relationship between natural resource rents and life expectancy at the country level, we also plot the relationship between predicted rents and 5-year changes in life expectancy for the four countries in SSA with the highest natural resource rents in the sample period: Angola, Congo Republic, Equatorial Guinea and Gabon.

### Estimation approach

To estimate empirical results on the relationship between change in life expectancy and natural resource rents, we use a conditional convergence framework [52, 53] using 5-year intervals between 1970 and 2015. The empirical model can be described as follows:

$$\ln(\Delta LE_{it,it-5}) = \alpha \ln(LE_{it-5}) + \beta \ln(GDP/cap_{it-5}) + \gamma \ln(Rents_{it-5}) + \theta X_{it} + a_i + b_i + \varepsilon_{it} \quad (1)$$

where $\gamma$ is the main coefficient of interest on *Rents*, and $\Delta$LE is the 5-year change in life expectancy in country $i$ between time $t$ and $t$-5. $LE_{it-5}$ and $GDP/cap_{it-5}$ are initial (beginning of the 5-year period) levels of life expectancy and income, respectively. $X_{it}$ is a k-dimension vector of time-varying control variables, including prevalence of HIV, percentage of births attended by skilled labour, urban population share, secondary school enrollment for females, tertiary school enrollment, total government revenue, government effectiveness, rule of law and control for corruption. $a_i$ and $b_i$ are the fixed effects for country and year, respectively. $\varepsilon_{it}$ is the error term.

In our 2SLS model, we use the energy price index to instrument for rents. Globally, commodity price tends to precede changes in general price level [54] and disproportionally affect countries heavily dependent on natural resource exports. Similar to previous work of [25, 55, 56], we control for country and year fixed effects in our estimation and explore country-year level changes in rents that result from the interaction between a country's average resource rents and global commodity prices. The main identifying assumption is that the price index is not correlated with any other factor driving changes in population health. The main logic of our model is that price shocks will affect all countries, but will disproportionally affect the rents of those countries that have the largest average natural resource endowments [43, 44, 55]. Given that most of the previous literature highlights IQ as a key moderator of natural resource effects, we also estimated separate models restricted to countries with high and low institutional quality; we also estimate separate models for SSA where political systems have been the most fragile over the past fifty years.

## Results

### Descriptive statistics

Table 1 presents unweighted summary statistics of data points across 186 countries over 5-year interval periods from 1970 to 2015. Mean life expectancy across all countries and all years was 65 years, with range values varying from 24 (Cambodia 1975) to 84 (Hong Kong SAR China 2015). The mean rent share in our sample was 7.88%, with a maximum value of 84.24% (Equatorial Guinea in 2000). The mean GDP per capita (measured in constant 2010 USD) over the sample period was USD 10,790.73 with a minimum of 137.60 (Mozambique 1985) and a

**Table 1. Descriptive statistics of the underlying indicators.**

| Variable | Obs | Coverage | Mean | Median | Std. Dev | Min | Max |
|---|---|---|---|---|---|---|---|
| Life expectancy | 1,842 | 1970 → 2015 | 64.78 | 67.72 | 10.81 | 23.60 | 84.28 |
| GDP/Capita | 1,555 | 1970 → 2015 | 10,791 | 3,458 | 16,351 | 138 | 141,200 |
| * Total rents | 1,592 | 1970 → 2015 | 7.88 | 2.61 | 12.10 | - | 84.24 |
| Population (in million) | 1,860 | 1970 → 2015 | 29.4m | 5.93m | 113m | 21,266 | 1,370m |
| % of urban population | 1,845 | 1970 → 2015 | 49.66 | 49.04 | 24.13 | 2.85 | 100.00 |
| Prevalence of HIV | 816 | 1970 → 2015 | 1.87 | 0.30 | 4.23 | 0.10 | 28.30 |
| School enrollment (sec. female) | 1,044 | 1970 → 2015 | 65.43 | 74.59 | 36.11 | 0.16 | 174.67 |
| School enrollment (tertiary) | 1,054 | 1970 → 2015 | 23.45 | 16.71 | 22.94 | - | 119.69 |
| Rule of law | 730 | 1996 → 2015 | (0.09) | (0.27) | 1.00 | (2.41) | 2.06 |
| Government effectiveness | 726 | 1996 → 2015 | (0.05) | (0.22) | 1.00 | (2.23) | 2.24 |
| Control for corruption | 730 | 1996 → 2015 | (0.06) | (0.33) | 1.01 | (1.77) | 2.44 |
| * Government revenue | 777 | 1970 → 2015 | 25.64 | 24.47 | 11.31 | 1.32 | 120.49 |
| Foreign direct investment | 1,394 | 1970 → 2015 | 3.23 | 1.54 | 7.04 | (25.78) | 103.34 |
| * Current health expenditure | 692 | 2000 → 2015 | 6.09 | 5.72 | 2.56 | 1.34 | 20.41 |

Notes: Data from World Bank Development Indicators (WDI). Data sampled from 186 countries. Data coverage varies across countries and across indicators.

* is measured in % GDP, m indicates that the figure is in millions. Parentheses denote a negative number.

maximum of 189,464.60 (Monaco 2015). The cross-sectional relationship between resource rents and life expectancy shows that life expectancy is negatively associated with an increase in resource rents (S2 Appendix).

## First stage regression and instrumental variable estimation

Table 2 presents the results from the first stage regressions for the five price indices considered. Overall, the energy index has strong prediction power of annual variation in total natural resource rents, with an F-statistic of 13.43.

Fig 1 shows the correlation between global energy indexes and the global average resource rents over the full 1970–2015 sample period (Fig 1). The correlation between average rents and the global energy index is 0.67, appearing to be stronger in the second half of the sample (post 1990).

## Estimation results

Table 3 shows the main ordinary least squares (OLS) and IV estimation results. Our preferred IV specification (Column 3) suggests that a 100% increase in resource rents results in a 2.7% increase in life expectancy (CI: 1%,4.4%). These results change only marginally when we add additional covariates in column 4. In terms of the covariates included, prevalence of HIV appears to be the only variable that consistently (and unsurprisingly) predicts subsequent changes in life expectancy.

Table 4 shows the results of the stratification by region and IQ. As shown in column 1 & 2, the treatment effects appear to be larger for SSA than for other regions. Impacts seem to be similar for countries with high and low IQ.

## Country case studies

Fig 2 shows average income growth as well as rents over the study period for the four countries with the largest natural resource rents (Fig 2). On average, in each of the four countries, total

**Table 2. First stage regression: Predicting rents with global price indices.**

| VARIABLES | Log (total rents, % GDP) | | | | |
|---|---|---|---|---|---|
| | **(1)** | **(2)** | **(3)** | **(4)** | **(5)** |
| Log (Pred. rents using *energy index*) | 0.144*** (0.039) | | | | |
| Log (Pred. rents using *non-energy index*) | | -0.104 (0.075) | | | |
| Log (Pred. rents using minerals index) | | | 0.148** (0.058) | | |
| Log (Pred. rents using metals index) | | | | 0.121** (0.059) | |
| Log (Pred. rents using precious metals index) | | | | | 0.081** (0.037) |
| Foreign Direct Investment, net inflows (% of GDP) | -0.000 (0.005) | -0.001 (0.005) | -0.001 (0.005) | -0.001 (0.005) | -0.001 (0.005) |
| Urban population (% of total population) | 0.006 (0.007) | 0.008 (0.007) | 0.008 (0.007) | 0.008 (0.007) | 0.007 (0.007) |
| Prevalence of HIV (% of population age 15–49) | -0.022 (0.021) | -0.020 (0.021) | -0.022 (0.021) | -0.021 (0.021) | -0.021 (0.021) |
| School enrollment, secondary. female (% Gross) | -0.007** (0.003) | -0.006* (0.003) | -0.007** (0.003) | -0.007** (0.003) | -0.007** (0.003) |
| School enrollment, tertiary (% Gross) | -0.010*** (0.004) | -0.013*** (0.004) | -0.012*** (0.004) | -0.012*** (0.004) | -0.012*** (0.004) |
| Constant | -0.787*** (0.241) | -0.858*** (0.243) | -0.866*** (0.241) | -0.875*** (0.241) | -0.836*** (0.241) |
| Observations | 1,836 | 1,836 | 1,836 | 1,836 | 1,836 |
| Number of countries | 186 | 186 | 186 | 186 | 186 |
| F-stats (instrument) | 13.45 | 1.94 | 6.50 | 4.15 | 4.66 |
| Prop > F | 0.0003 | 0.1638 | 0.0108 | 0.0416 | 0.0310 |

Notes: Standard errors in parentheses,

*** $p < 0.01$,

** $p < 0.05$,

* $p < 0.1$.

Notes: All models include country and year fixed effects. Coefficients displayed are linear regression coefficients with cluster-robust standard errors in parentheses.

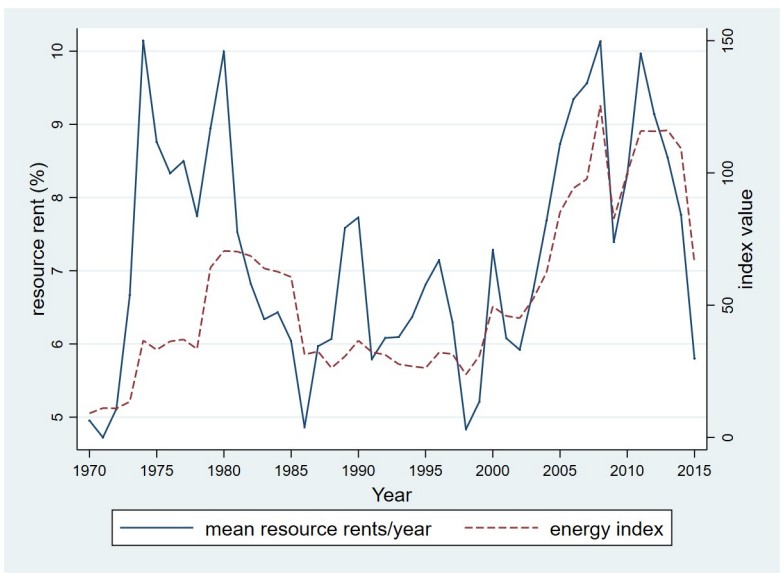

**Fig 1. Relationship between energy index and resource rents.** Data source: World Development Indicators (WDI) and The World Bank Commodity Price Data (The Pink sheet).

**Table 3. Causal impact of rents on changes in life expectancy.**

| VARIABLES | [Log (Change in life expectancy)]$_{t,t-5}$ | | | |
| --- | --- | --- | --- | --- |
| | OLS | | IV | |
| | (1) | (2) | (3) | (4) |
| [Log (total rents)]$_{t-5}$ | 0.001 (0.001) | 0.001 (0.001) | 0.027*** (0.009) | 0.028*** (0.010) |
| Log (life expectancy)$_{t-5}$ | -0.218*** (0.036) | -0.247*** (0.045) | -0.240*** (0.040) | -0.255*** (0.049) |
| Log (GDP/Cap)$_{t-5}$ | -0.012*** (0.005) | -0.009* (0.005) | -0.003 (0.006) | -0.001 (0.007) |
| [Foreign Direct Investment, net inflows (% of GDP)]$_t$ | | 0.000 (0.000) | | 0.000 (0.000) |
| [Urban population (% of total population)]$_t$ | | 0.001* (0.000) | | 0.000 (0.000) |
| [Prevalence of HIV (% of population ages 15–49)]$_t$ | | -0.003** (0.002) | | -0.003* (0.002) |
| [School enrollment, sec. female (% Gross)]$_t$ | | -0.000 (0.000) | | -0.000 (0.000) |
| [School enrollment, tertiary (% Gross)]$_t$ | | -0.000 (0.000) | | -0.000 (0.000) |
| Constant | 0.960*** (0.141) | 1.041*** (0.163) | 1.006*** (0.151) | 1.048*** (0.174) |
| Observations | 1,656 | 1,656 | 1,656 | 1,656 |
| Number of country | 186 | 186 | 186 | 186 |

Notes: All models include country and year fixed effects.

Cluster-robust standard errors in parentheses,

*** $p<0.01$,

** $p<0.05$,

* $p<0.1$.

natural resource rents contributed over 20% of the national GDP, with Equatorial Guinea having the highest average of 30.44% (Fig 2). Global natural resource and energy prices were highest in the 1970 and 1980s and lowest in the 1990s and early 2000s. Angola and Equitorial Guinea experienced positive changes in life expectancy throughout the period, with the largest improvements reported in Angola in 2005 and 2010. Overall, the patterns seen in Angola, Republic of Congo and Gabon seem well aligned with the main results presented in the previous section: relatively steady improvements in the 1970s and 1980s as well as post 2005, when resource prices were high, and much weaker (or negative) improvements in the 1990s when resource prices were low. Even though the weaker performance during the 1990s was at least partially due to HIV [57], none of the countries shown here was hit particularly hard by HIV, with HIV prevalence rates below 5% throughout the sample period [58]. The most noticeable outlier is Equatorial Guinea: even though the country very heavily depends on its oil exports, it seems to have succeeded in maintaining steady increases in life expectancy.

## Discussion

In this study, we use a large cross-country panel dataset to assess the causal effects of natural resource rents on changes in population health during the period 1970 to 2015 using instrumental variable regression. In a first step, we show that global variations in commodity prices strongly predict the magnitude of natural resource rents earned by countries conditional on country and year fixed effects. Using the observed variation in rents created by global prices in our models, we then show that each standard deviation increase in resource rents results in about one additional year of life expectancy over a five-year period.

While we find no evidence of short-term changes in economic growth, we find that fluctuations in global commodity prices do result in substantial changes in government revenue (Table 5). These results are similar to Deaton and Miller [59] and Collier et al. [60] who use vector autoregressive models to test the effects of commodity prices on short-term incomes. It

**Table 4. Impact of rents by region, institutional quality and current health expenditure.**

| VARIABLES | Log (Change in life expectancy)$_{t,t-5}$ | | | | | |
|---|---|---|---|---|---|---|
| | Excluding SSA | SSA | Low Institutional Quality | High Institutional Quality | Low Current Health Expenditure | High Current Health Expenditure |
| Log(total rents)$_{t-5}$ | 0.003 (0.006) | 0.062 (14.477) | 0.043* (0.025) | 0.013 (0.012) | 0.010 (0.009) | 0.058* (0.032) |
| Log(life expectancy)$_{t-5}$ | -0.234*** (0.068) | -0.288 (0.966) | -0.257*** (0.063) | -0.245*** (0.059) | -0.168*** (0.028) | -0.319*** (0.077) |
| Log(GDP/Cap)$_{t-5}$ | -0.004 (0.006) | -0.012 (0.343) | -0.010 (0.009) | 0.007 (0.008) | -0.005 (0.005) | 0.019 (0.028) |
| [Foreign Direct Investment, net inflows (% of GDP)]$_t$ | 0.000 (0.000) | 0.000 (0.014) | 0.000 (0.000) | 0.000 (0.000) | 0.000 (0.000) | 0.000 (0.000) |
| [Urban population (% of total population)]$_t$ | 0.001 (0.001) | -0.000 (0.217) | 0.001 (0.001) | 0.000 (0.000) | 0.000 (0.000) | 0.001 (0.001) |
| [Prevalence of HIV (% of population ages 15–49)]$_t$ | -0.000 (0.004) | -0.002 (0.705) | -0.003 (0.002) | -0.002** (0.001) | -0.001 (0.002) | -0.004* (0.002) |
| [School enrollment, sec. female (% Gross)]$_t$ | -0.000 (0.000) | -0.000 (0.109) | -0.000 (0.000) | -0.000 (0.000) | -0.000 (0.000) | -0.000 (0.000) |
| [School enrollment, tertiary (% Gross)]$_t$ | -0.000 (0.000) | 0.001 (0.078) | -0.000 (0.001) | 0.000 (0.000) | 0.000 (0.000) | -0.000 (0.000) |
| Constant | 0.965*** (0.242) | 1.060 (30.384) | 1.120*** (0.219) | 1.074*** (0.256) | 0.744*** (0.119) | 1.178*** (0.273) |
| Observations | 1,226 | 430 | 1,026 | 630 | 880 | 776 |
| Number of countries | 138 | 48 | 115 | 71 | 98 | 88 |

Cluster-robust standard errors in parentheses,

*** p<0.01,

** p<0.05,

* p<0.1.

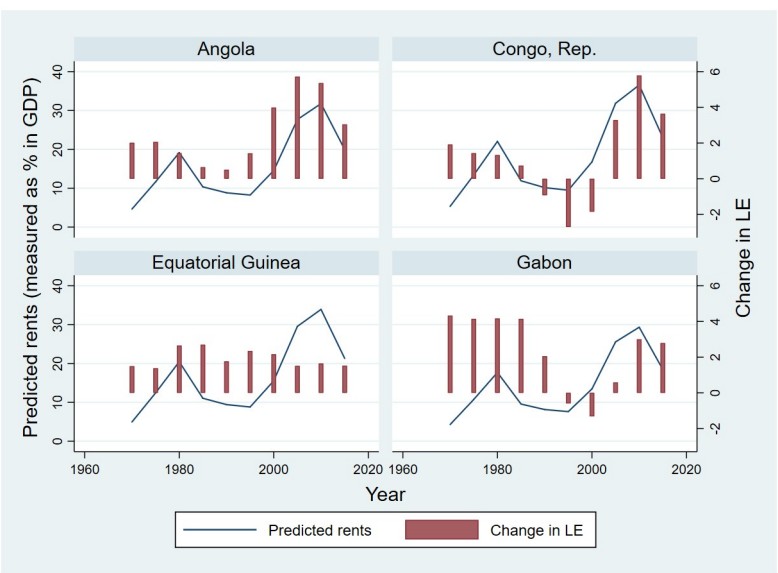

**Fig 2. Relationship between predicted rents and change in life expectancy in five-year intervals.** Data source: World Development Indicators (WDI).

**Table 5. Relationship between resource rents and government revenue and health expenditure.**

| VARIABLES | Log (Revenue) | Log(Current Health Expenditure) | Log(GDP/Cap) |
|---|---|---|---|
| Log(total rents) | 0.165* (0.096) | -0.052 (0.073) | -0.087 (0.104) |
| Log(life expectancy) | 0.400* (0.209) | 0.176 (0.157) | 0.307 (0.234) |
| Log(GDP/Cap) | 0.082 (0.056) | -0.080 (0.055) | |
| Foreign Direct Investment, net inflows (% of GDP) | 0.001 (0.001) | -0.001 (0.001) | 0.000 (0.002) |
| Urban population (% of total population) | 0.003 (0.003) | -0.008** (0.004) | 0.009* (0.005) |
| Prevalence of HIV (% of population ages 15–49) | 0.008 (0.009) | -0.004 (0.008) | 0.005 (0.010) |
| School enrollment, Sec. Female (% Gross) | 0.002 (0.001) | 0.000 (0.001) | 0.004** (0.001) |
| School enrollment, tertiary (% Gross) | 0.002 (0.002) | 0.001 (0.001) | 0.008*** (0.002) |
| Constant | 0.195 (0.806) | 2.104*** (0.675) | 4.531*** (0.896) |
| Observations | 1,842 | 1,842 | 1,842 |
| Number of countries | 186 | 186 | 186 |

Cluster-robust standard errors in parentheses,

*** $p<0.01$,

** $p<0.05$,

* $p<0.1$

seems plausible that changes in government revenue increase governments' ability to support public health and development programs that can potentially contribute to improved population health. While our data does not allow us to directly identify the causal mechanisms driving these differences, one possible explanation for the larger effects of rents observed in SSA is that governments in stable settings (i.e. with strong institutions and fiscal management) may be better able to smooth incomes over time either by building up reserves (e.g. Norway) [61], through increased borrowing in periods when commodity prices are low, or by hedging resource prices in global markets [62]. This will, however, be difficult in countries with limited fiscal discipline and weak institutions, which will therefore be more exposed to cyclical revenue streams as well as cyclical changes in life expectancy, as shown in our subsample analysis. When we stratify our sample by major geographical regions and average institutional quality, we find large positive effects in the SSA sample, even though the differences across subgroups are not statistically significant. We also observe on average larger resource effects on countries with lower institutional quality; however, estimates on countries with lower institutional quality are relatively imprecise so that subgroup confidence intervals overlap.

A recent scoping review on health in the context of resource extraction suggests that most studies on the relationship between natural resources and health focus on occupational health risks and exposures to toxic substances related to mining activities [31]. The few studies available to date investigating community-level health impacts mostly focus on specific health conditions, highlighting negative effects of mining on malnutrition, malaria, HIV and mental health in specific settings [34, 35, 37]. The results presented in this study are conceptually different from these previous studies because we focus on average country-level health outcomes rather than health outcomes directly observed in contexts where resources are extracted. Even though extractive industries may contribute to improved health outcomes locally through improved public infrastructures, employment and business opportunities, these effects are likely to be too small to be reflected in national estimates due to the relatively small populations directly exposed to such projects. From a central government perspective, natural resources are primarily of interest as a source of additional income, allowing governments to promote access to and improve quality of health services, along with socio-economic development

more broadly [63, 64] that can contribute to improved health outcomes [41, 64, 65]. Our results suggest that booms in global commodity prices do indeed create windfall revenue gains for governments in the short run. This does not imply, however, that natural resources are necessarily positive for health or development in the long run: increased revenues during booms also imply negative revenue shocks during global commodity market contractions, with likely immediate negative repercussions on government funding to social and health programs unless these shocks can be offset by fiscal reserves or external financing [66].

### Strengths and limitations

This study is, to our knowledge, the most comprehensive analysis of the empirical relationship between natural resources and improvements in life expectancy using cross-country data to date. Despite the large data set compiled for this study, several limitations are worth highlighting. First, and as already mentioned, missing data is quite common in the WDI database. This was addressed by limiting data extraction to: (1) a specific set of indicators relating to population, health and resource endowment; (2) restricting to data between 1970 and 2017; and (3) by using a multiple imputation algorithm. A second limitation of all cross-country analyses is that variables capturing complex aspects of social, political and health systems are largely lacking. These factors are likely important for understanding some of the changes in life expectancy observed and may help explain the empirical relationships seen in our analysis. Our analysis also focuses on variation in rents that are driven by energy prices—fluctuations in other rents may affect countries in different ways not captured in our analysis. Finally, we did not adjust for geopolitical or global events such as wars and disease epidemics. While these events should not be correlated with the price series conditional on country and year fixed effects, we cannot fully rule out residual confounding concerns.

### Conclusion

The results presented in this paper suggest that natural resources can help governments in low- and middle-income countries to improve population health. On average, we find that life expectancy in countries with large natural resource endowments improves more rapidly than life expectancy of countries with small endowments during the peirod with high commodity prices. However, this also implies that global contractions in commodity prices can slow down progress in population health. To avoid these negative repercussions during commodity price contractions, countries with large resource endowments should put mechanisms in place that allow smoothing of commodity-related incomes over time, and ensure sustainable and continued financing for public services, including health care. More research is needed to identify the best mechanisms for countries to reach this goal.

### Supporting information

**S1 Appendix. Indicator definitions.**
(DOCX)

**S2 Appendix. Short-run correlation matrix (multiple observation per country).**
(DOCX)

### Acknowledgments

The authors would like to thank the entire team of the Health Impact Assessment for Sustainable Development (HIA4SD) Project both at the Swiss Tropical and Public Health Institute

and partner institutions in the project countries for their continuous support over the course of this study.

## Author Contributions

**Conceptualization:** Isaac Lyatuu, Günther Fink.

**Formal analysis:** Isaac Lyatuu, Günther Fink.

**Funding acquisition:** Mirko S. Winkler.

**Methodology:** Günther Fink.

**Supervision:** Mirko S. Winkler, Günther Fink.

**Writing – original draft:** Isaac Lyatuu.

**Writing – review & editing:** Isaac Lyatuu, Georg Loss, Andrea Farnham, Mirko S. Winkler, Günther Fink.

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
