## [Decision Letter · Decision Letter 0]

20 Apr 2021

PONE-D-21-08827

Short-term effects of national-level natural resource rents on life expectancy: A cross-country panel data analysis

PLOS ONE

Dear Dr. Isaac Lyatuu,

Thank you for submitting your manuscript to PLOS ONE. After careful consideration, we feel that it has merit but does not fully meet PLOS ONE’s publication criteria as it currently stands. Therefore, we invite you to submit a revised version of the manuscript that addresses the points raised during the review process.

We look forward to receiving your revised manuscript.

Kind regards,

László VASA, PhD

Academic Editor

PLOS ONE

Journal Requirements:

Thank you for stating the following in the Acknowledgments Section of your manuscript:

The authors would like to thank the Swiss Programme for Research on Global Issues for

Development (r4d Programme), which is a joint funding initiative by the Swiss Agency for

Development and Cooperation (SDC) and the Swiss National Science Foundation (SNSF) [grant

number 169461] for having financed this study.

The funders had no role in study design, data collection and analysis, decision to publish, or preparation of the manuscript

Reviewers' comments:

Reviewer's Responses to Questions

**Comments to the Author**

1. Is the manuscript technically sound, and do the data support the conclusions?

Reviewer #1: Partly

Reviewer #2: Yes

2. Has the statistical analysis been performed appropriately and rigorously? 

Reviewer #1: Yes

Reviewer #2: Yes

3. Have the authors made all data underlying the findings in their manuscript fully available?

Reviewer #1: Yes

Reviewer #2: Yes

4. Is the manuscript presented in an intelligible fashion and written in standard English?

Reviewer #1: Yes

Reviewer #2: Yes

5. Review Comments to the Author

Reviewer #1: The topic of the paper is unique, the level of methodology and evaluation is high, it is a very promising paper, however:

- introduction is very short and not appropriate regarding the content,

- there is absolutely no literature review in the paper,

- context isn't highlighted well,

- very short article as a whole.

Reviewer #2: The topic is extremely interesting and inspiring. I consider it a very good idea to study the relationship between natural resources and human health. I also consider the database and the method used to be suitable for the study. Although the range of data processed can be identified, summarizing the categories in a table would improve clarity.

What I find problematic is that neither the title nor the abstract show that the study applies to some African countries. This may even be good, but it’s worth clarifying when positioning the article.

The conclusion needs to be completed.

Editing errors need to be corrected (Table 1 hangs from the page.)

6. PLOS authors have the option to publish the peer review history of their article (what does this mean?). If published, this will include your full peer review and any attached files.

Reviewer #1: No

Reviewer #2: No

---

## [Author Response · Author response to Decision Letter 0]

5 May 2021

Point-by-point response, 

Research article: PONE-D-21-08827

Kindly find point by point responses to reviewer 1 & 2. In addition, we have attached a word document with these feedback. 

Reviewer #1

1. Is the manuscript technically sound, and do the data support the conclusions?

Reviewer #1: Partly

Response: We thank Reviewer #1 for having scrutinized our manuscript and are grateful for the recommendations received. We hope that the reviewer is now fully satisfied with the quality of the paper after having addressed the reviewer’s concerns as specified below.

2. Has the statistical analysis been performed appropriately and rigorously?

Reviewer #1: Yes

Response: Thank you.

3. Have the authors made all data underlying the findings in their manuscript fully available?

Reviewer #1: Yes

Response: Thank you.

4. Is the manuscript presented in an intelligible fashion and written in standard English?

Reviewer #1: Yes

Response: Thank you.

5. Review Comments to the Author:

The topic of the paper is unique, the level of methodology and evaluation is high, it is a very promising paper, however:

- introduction is very short and not appropriate regarding the content,

- there is absolutely no literature review in the paper,

- context isn't highlighted well,

- very short article as a whole.

Response: We are pleased about the overall very positive appraisal of our study. In response to the reviewer’s request to provide more background while better embedding the paper in existing literature on the topic, we have now expanded the introduction for providing more background (see lines 31-37, 58-69, 79-82 in the revised manuscript), which also includes additional references. Even though we do not have a formal literature section, we summarize the main literature on economic growth and the resource curse in the first two paragraphs for the introduction, and then provide an overview of the health-focused literature in lines (58-69) of the revised manuscript. We would be happy to move this part of the introduction to a separate “Literature review” section if this is preferred by the Editor and reviewer.

Moreover, we have also expanded the conclusion chapter of the paper (see lines 298-302).

We hope that these additions have extended our work to the level that is sufficient. The overall article length is now just below 4000 words, which seems well aligned with word limits of most general interest journals.

 

Reviewer #2

1. Is the manuscript technically sound, and do the data support the conclusions?

Reviewer #2: Yes

Response: We thank Reviewer #2 for having scrutinized our manuscript and are grateful for the recommendations received.

2. Has the statistical analysis been performed appropriately and rigorously?

Reviewer #2: Yes

Response: Thank you.

3. Have the authors made all data underlying the findings in their manuscript fully available?

Reviewer #2: Yes

Response: Thank you.

4. Is the manuscript presented in an intelligible fashion and written in standard English?

Reviewer #2: Yes

Response: Thank you.

5. Review Comments to the Author:

The topic is extremely interesting and inspiring. I consider it a very good idea to study the relationship between natural resources and human health. I also consider the database and the method used to be suitable for the study. Although the range of data processed can be identified, summarizing the categories in a table would improve clarity.

Response: We are grateful for the overall positive evaluation of our study by Reviewer #2. We are sorry that the prior formatting of Table 1 (which has been fixed now) did not allow to see all the detail of the data. The full table contains means, standard deviation, min and max values as well as the temporal range, and should thus give readers a clear sense of the data used. We would of course be delighted to add further information or statistics if there are specific suggestions or requests.

What I find problematic is that neither the title nor the abstract show that the study applies to some African countries. This may even be good, but it’s worth clarifying when positioning the article.

Response: Our manuscript does indeed have an additional layer of analysis focusing on the African continent. In order to be more consistent on this aspect, as requested by Reviewer #2, the introduction now includes a few sentences that clarify why the study presented is of particular relevance for the African continent (lines 31-37) and in the abstract we added an Africa-specific finding (lines 23-24). We also clarified at the end of the introduction that we look both at Africa and the rest of the developing world (lines 79-82). But we prefer not to include “Africa” in the title as we primarily present a global analysis, with Africa as a secondary, supplementary layer of analysis.

The conclusion needs to be completed.

Response: We have further expanded the conclusion of the paper (lines 298-302).

Editing errors need to be corrected (Table 1 hangs from the page.)

Response: We have once more read our manuscript sentence-by-sentence and made corrections where needed. Also the layout of Table 1 was corrected.

 

Journal Requirements:

Response: We have carefully studied the PLOS ONE's style requirements and applied them when developing the manuscript.

The authors would like to thank the Swiss Programme for Research on Global Issues for Development (r4d Programme), which is a joint funding initiative by the Swiss Agency for Development and Cooperation (SDC) and the Swiss National Science Foundation (SNSF) [grantnumber 169461] for having financed this study.

The funders had no role in study design, data collection and analysis, decision to publish, or preparation of the manuscript

Response: Many thanks for the clarification of the difference between the Funding Statement and the acknowledgements with PLOS ONE.

We have replaced the previous content of the acknowledgement section: “The authors would like to thank the entire team of the Health Impact Assessment for Sustainable Development (HIA4SD) Project both at the Swiss Tropical and Public Health Institute and partner institutions in the project countries for their continuous support over the course of this study” (lines 308-310).

Kindly include the following content in the Funding Statement: “The authors would like to acknowledge the Swiss Programme for Research on Global Issues for Development (r4d Programme), which is a joint funding initiative by the Swiss Agency for Development and Cooperation (SDC) and the Swiss National Science Foundation (SNSF). The funders had no role in study design, data collection and analysis, decision to publish, or preparation of the manuscript.”

---

## [Decision Letter · Decision Letter 1]

9 May 2021

PONE-D-21-08827R1

Short-term effects of national-level natural resource rents on life expectancy: A cross-country panel data analysis

PLOS ONE

Dear Dr. Isaac Lyatuu,

Thank you for submitting your manuscript to PLOS ONE. After careful consideration, we feel that it has merit but does not fully meet PLOS ONE’s publication criteria as it currently stands. Therefore, we invite you to submit a revised version of the manuscript that addresses the points raised during the review process.

We look forward to receiving your revised manuscript.

Kind regards,

László VASA, PhD

Academic Editor

PLOS ONE

Journal Requirements:

Reviewers' comments:

Reviewer's Responses to Questions

**Comments to the Author**

1. If the authors have adequately addressed your comments raised in a previous round of review and you feel that this manuscript is now acceptable for publication, you may indicate that here to bypass the “Comments to the Author” section, enter your conflict of interest statement in the “Confidential to Editor” section, and submit your "Accept" recommendation.

Reviewer #1: All comments have been addressed

2. Is the manuscript technically sound, and do the data support the conclusions?

Reviewer #1: Partly

3. Has the statistical analysis been performed appropriately and rigorously? 

Reviewer #1: Yes

4. Have the authors made all data underlying the findings in their manuscript fully available?

Reviewer #1: Yes

5. Is the manuscript presented in an intelligible fashion and written in standard English?

Reviewer #1: Yes

6. Review Comments to the Author

Reviewer #1: The authors mostly accepted my recommendation in the previous review process and did the necessary improvements. However, regarding the literature review, it is not enough what they did. I recommend to formulate a separate literature review chapter and extending the sources to be processed.

7. PLOS authors have the option to publish the peer review history of their article (what does this mean?). If published, this will include your full peer review and any attached files.

Reviewer #1: No

---

## [Author Response · Author response to Decision Letter 1]

9 May 2021

Dear editors, 

Once again, thank you for the opportunity to revise our submission. We have uploaded, 

1. Response to reviewers document

2. Revised manuscript with track changes

3. Manuscript (revised)

Kind regards,

Isaac Lyatuu

---

## [Editor Report · Decision Letter 2]

14 May 2021

Short-term effects of national-level natural resource rents on life expectancy: A cross-country panel data analysis

PONE-D-21-08827R2

Dear Dr. Isaac Lyatuu,

We’re pleased to inform you that your manuscript has been judged scientifically suitable for publication and will be formally accepted for publication once it meets all outstanding technical requirements.

Kind regards,

László VASA, PhD

Academic Editor

PLOS ONE
---

## [Editor Report · Acceptance letter]

21 May 2021

PONE-D-21-08827R2 

Short-term effects of national-level natural resource rents on life expectancy: A cross-country panel data analysis 

Dear Dr. Lyatuu:

I'm pleased to inform you that your manuscript has been deemed suitable for publication in PLOS ONE. Congratulations! Your manuscript is now with our production department. 

Kind regards, 

on behalf of

Prof. Dr. László Vasa 

Academic Editor

PLOS ONE